# Recent Advances in Gold(I)-Catalyzed Approaches to Three-Type Small-Molecule Scaffolds via Arylalkyne Activation

**DOI:** 10.3390/molecules27248956

**Published:** 2022-12-15

**Authors:** Lu Yang, Hongwei Su, Yue Sun, Sen Zhang, Maosheng Cheng, Yongxiang Liu

**Affiliations:** 1Key Laboratory of Structure-Based Drug Design and Discovery of Ministry of Education, Shenyang Pharmaceutical University, Shenyang 110016, China; 2Wuya College of Innovation, Shenyang Pharmaceutical University, Shenyang 110016, China; 3Institute of Drug Research in Medicine Capital of China, Shenyang Pharmaceutical University, Benxi 117000, China

**Keywords:** gold(I)-catalyzed, arylalkyne, benzene derivatives, cyclopentene derivatives, furan and pyran derivatives

## Abstract

Gold catalysts possess the advantages of water and oxygen resistance, with the possibility of catalyzing many novel chemical transformations, especially in the syntheses of small-molecule skeletons, in addition to achieving the rapid construction of multiple chemical bonds and ring systems in one step. In this feature paper, we summarize recent advances in the construction of small-molecule scaffolds, such as benzene, cyclopentene, furan, and pyran, based on gold-catalyzed cyclization of arylalkyne derivatives within the last decade. We hope that this review will serve as a useful reference for chemists to apply gold-catalyzed strategies to the syntheses of related natural products and active molecules, hopefully providing useful guidance for the exploration of additional novel gold-catalyzed approaches.

## 1. Introduction

Gold was long considered an inert precious metal that cannot be used in catalyzing chemical reactions until Bond and Ito discovered that gold exhibits excellent catalytic activity in nanoparticle form or as soluble complexes [1,2], opening the door for the subsequent development and application of gold-catalyzed chemical reactions [3]. The oxidation states of gold include Au(0), Au(I), and Au(III). Au(I) alone is unstable in solution and is generally used in linear complexes with phosphine ligands, carbene ligands, etc. (Figure 1a) [4]. The counterions of gold catalysts include trifluoromethanesulfonate (OTf^−^), tetrafluoroborate (BF_4_^−^), hexafluoroantimonate (SbF_6_^−^), tetraphenylboron (BAr_4_^−^), etc. (Figure 1b). A reactive Au(I) complex is formed through counterion exchange with various silver salts (AgX) or with sodium tetra-aryl borate (NaBAr_4_) and potassium tetra-aryl borate (KBAr_4_) (Figure 1c).

In homogeneous gold-catalyzed reactions, gold, as a soft acid, is highly nucleophilic to the π-electron system in alkynes, alkenes, and allenes, promoting a series of chemical transformations. In 1998, the Teles group first reported the hydrofunctionalization of alkynes by a Au(I)-phosphine complex, after which the great potential of homogeneous gold catalysis in organic synthesis was gradually explored [5]. Over the past two decades, many subtle gold-catalyzed methodologies have been developed, including cycloaddition reactions, cycloisomerization reactions, and cascade cyclization reactions.

Gold catalysts are characterized by high catalytic reactivity, good chemical selectivity, mild reaction conditions, and high tolerance to water and air. The most common application of gold catalysts in organic synthesis is the cyclization reaction, which can be used to synthesize a benzene ring, indole ring, quinoline ring, imidazole ring, oxazole ring, etc. [6,7,8,9,10,11,12,13,14,15,16,17,18,19,20,21]. Arylalkyne-containing building blocks are easily prepared and can undergo a variety of cyclization reactions, offering unique advantages with respect to the construction of small-molecule skeletons, such as benzenes, cyclopentenes, furans, and pyrans under the influence of gold catalysis (Figure 2). Therefore, we attempted to systematically summarize the building-block-directed construction of specific small-molecule scaffolds with arylalkyne substrates under gold(I)-catalyzed conditions within the last decade, and any works missed were unintentional.

In this feature paper, studies are classified according to the structural characteristics of small-molecule skeletons, highlighting the development of strategies and the scope of research on gold-catalyzed cyclization of arylalkyne derivatives, including arene–diynes, arene–enynes, aryne–enolether, aryne–acetals, etc.

## 2. Syntheses of Benzene Derivatives

Many important natural products and drugs contain aromatic units, such as benzenes, naphthalenes, and biaryls; thus, the construction of benzene rings is significant in organic synthesis. The synthesis methods for benzene rings using [2+2+2] or [4+2] cycloaddition reactions usually require harsh conditions. However, Au(I)-catalyzed cyclization of arylalkyne substrates represents a mild and versatile approach to the construction of benzene rings. In this chapter, we summarize previous works on the synthesis of benzene derivatives based on the type of arylalkyne.

### 2.1. Arene-Diyne Substrates

In 2012, Hashmi and colleagues reported a double gold(I)-activated cyclization of arene-diynes to construct benzene rings for the synthesis of *β*-substituted naphthalene derivatives, which was achieved through an unexpected reaction pathway (Figure 1) [22]. First, one terminal alkyne of arene-diyne (**1**) was activated by a gold catalyst to form a Au–C-σ bond through catalyst transfer, and the other terminal alkyne was activated to produce a double-activated intermediate (**2**). Subsequently, the activated triple bond was attacked by the *β* carbon of gold acetylide due to π coordination, which induced the formation of a five-membered ring to generate gold–vinylidene (**3**). Next, intermediate **4** was formed by a solvent attack (benzene) and a *1,3-H* shift, which was subsequently transformed into intermediate **5** via a ring expansion. Finally, after the elimination of the gold(I) catalyst and protonation, a *β*-substituted naphthalene product (**6**) was obtained. The reaction pathway was clearly verified through X-ray crystal structure analysis of the key intermediates and controlled experiments. The strategy of double gold activation had a significant influence on the later development of gold chemistry. 

In the same year, the Ohno group described a gold(I)-catalyzed tandem approach to 1,3-disubstituted naphthalenes using arene-diynes with 14 examples, achieving a quantitative yield (Figure 2) [23]. This strategy mainly involves intermolecular nucleophilic addition and intramolecular nucleophilic addition reactions. Gold(I)-activated terminal alkyne was first attacked by nucleophilic reagents, such as ROH, RR’NH, and Ar-H, to generate intermediates (**8**) that were immediately converted to enolether/enamine-type intermediates (**9**) by protodeauration. A subsequent 6-*endo*-dig cyclization yielded intermediates (**10**) that underwent subsequent aromatization and protonation to provide naphthalene derivatives (**11**). The above reaction path was verified in detail by the syntheses of silyl enolether intermediates (**9**) and related deuteration experiments.

### 2.2. Arene-Enyne Substrates

Gold-catalyzed cyclization of arene-enynes is an important strategy for building small-molecule carbocyclic skeletons that has inspired many excellent methods to be reported. In 2017, Shi et al., developed a gold(I)-catalyzed tandem cyclization–oxidation strategy to access aryl acetaldehyde derivatives using alkylidene–cyclopropane and pyridine *N*-oxide (Figure 3) [24]. First, coordination of the triple bond by [Au]^+^ triggered the 6-*endo*-dig cyclization to form intermediates (**13**), and benzylic carbocation was stabilized by electron-rich cyclopropane and the benzene ring. Subsequently, the 3,5-dibromo-pyridine *N*-oxide acted as a nucleophile to attack cyclopropane and produce intermediates (**14**). Finally, aryl acetaldehyde derivatives (**15**) were generated by Kornblum-type oxidation with the simultaneous release of 3,5-dibromo-pyridine. The scope of application of the above strategy was examined using 27 examples with 36–93% yields. It is worth noting that when R^2^ was a substituted phenyl group, the aryl acetaldehyde derivative could be further modified to polycyclic aromatic hydrocarbons (PAHs) under the catalysis of In(OTf)_3_.

In 2019, the Ohno lab described a gold(I)-catalyzed cascade cyclization strategy for the syntheses of cyclopropanes derivatives, with 11 examples and yields of up to 96% (Figure 4) [25]. The activation of the allenyl moiety of 1-allenyl-2-ethynyl-3-alkylbenzene substrates (**16**) by the gold complex induced a nucleophilic attack of alkyne to yield vinyl cationic intermediates (**17**). Then, a *1,5-H* shift occurred to generate benzylic carbocation intermediates (**18**). Subsequent carbocation cyclization provided acenaphthene derivatives (**19**) after aromatization and protodeauration. In addition, a series of 1-(naphth-1-yl)cyclopropa-[*b*]benzofuran derivatives was successfully prepared when phenylene-tethered allenynes and benzofurans were subjected to the same gold-catalyzed conditions.

In 2021, the Ohno lab reported the syntheses of benzo[*cd*]indole skeletons by gold-catalyzed tandem cyclization based on their previous work (Figure 5) [26]. In this approach, a series of amino-allenyne substrates (**20**) were designed and prepared. First, the activated allene was attacked by the electron-rich alkyne to form vinyl cationic intermediates (**21**). The vinyl cation was captured by the neighboring amine group to yield tricyclic fused indoles (**22**), which underwent an isomerization to furnish pyrrolonaphthalenes (**23**). The resulting tricyclic derivatives could be transformed into nitrogen-containing polycyclic aromatic compounds (*N*-PACs) with special photophysical properties through *N*-arylation or Friedel–Crafts acylation.

Recently, Liu and colleagues achieved the gold(I)-catalyzed construction of benzene derivatives using arene–enyne substrates, which was applied to the total syntheses of eight natural products (Figure 6) [27]. Coordination of alkyne in substrates (**24**) promoted a 6-*endo*-dig cyclization to yield intermediates (**25**) that were converted into iodonaphthalenes (**26**) in situ in the presence of *N*-iodosuccinimide (NIS). The intermediates (**26**) were used as key moieties to synthesize benzo[*c*]phenanthridine alkaloids in a pot-economic approach. Moreover, the cytotoxicities of these alkaloids were investigated, indicating the future potential of these molecules for anticancer research.

### 2.3. Aryne-Enolether Substrates

Enolether showed versatile properties in gold-catalyzed reactions, making it suitable for use not only as a nucleophile but also as an electrophile to be coordinated by [Au]^+^. The combination of enolether with alkyne derivatives to form building blocks containing 1,5-enyne showed unique advantages in gold-catalyzed tandem cyclization for the syntheses of benzene derivatives. Accordingly, the Liu lab has reported a number of such studies in recent years.

In 2014, a gold(I)-catalyzed cycloisomerization of arylalkyne-enolether for the construction of multisubstituted naphthalenes was developed by the Liu group (Figure 7) [28]. First, the triple bonds of the substrates (**27**) were activated by the gold species, which induced an intermolecular nucleophilic addition by alcohol to yield dienol ether intermediates (**28**). Coordination of [Au]^+^ to the electron-rich enolether promoted cycloisomerization to provide multisubstituted naphthalenes (**29**) via the release of methanol and protodeauration. The scope of this strategy was examined by synthesizing 20 alkyne–enolether substrates with 38–88% yields.

In 2017, Liu and colleagues achieved gold(I)-catalyzed tandem cyclization for the syntheses of benzo[*a*]carbazole derivatives using arylalkyne-enolether substrates (Figure 8) [29]. The authors modulated the electronic properties of the triple bond through the substituent of the right benzene ring, which further tuned the cyclization order. When there were sulfonamide substituents on the appropriate benzenes, the *α*-position of the alkyne activated by [Au]^+^ induced a 5-*endo*-dig cyclization to produce indole intermediates (**31**). The enolether was then activated by a gold(I) complex and attacked by the electron-rich indole to promote the second cyclization, yielding benzo[*a*]carbazoles (**32**) by elimination of methanol and protodeauration. The above reaction mechanism was verified by capturing the intermediates and further supported by DFT calculations. Notably, when the appropriate benzene rings of the substrates were substituted with amine groups, the order of cyclization was changed to yield indeno-[1,2-*c*]quinoline derivatives, which are described in detail in later sections.

One year later, another cascade cyclization strategy was reported by the Liu group as an ongoing study on the gold-catalyzed cyclization of enolether-involved substrates in the construction of small-molecule scaffolds. The authors achieved the syntheses of xanthone and acridone derivatives by designing a series of alkyne–enolether substrates with 25 examples and up to 98% yield (Figure 9) [30]. Initially, the triple bonds of the substrates (**33**) were chelated by the gold(I) species, which promoted an intramolecular Michael addition to obtain intermediates (**34**) after protodeauration. Then, gold(I)-activated enolether was attacked by newly generated enolethers or enamines to undergo a 6-*endo*-trig cyclization. Finally, xanthone or acridone derivatives (**35**) were achieved via a similar pathway as reported previously.

In 2022, Liu and colleagues reported a gold(I)-catalyzed 6-*endo*-dig cyclization of arylalkyne–enolethers (**36**) to construct 2-(naphthalen-2-yl)aniline derivatives (Figure 10) [31]. The authors found that the amine group on the right-hand benzene ring benefited 6-*endo*-dig cyclization via an electron-donating effect to generate naphthalenes (**37**) after isomerization and protodeauration. In addition, several important heterocycles (**38–41**) were synthesized in a divergent manner from that of naphthalene derivatives (**37**).

### 2.4. Other Arylalkyne Substrates

In 2013, Ye et al., described a gold(I)/acid-catalyzed methodology for the syntheses of anthracenes using *o*-alkynyl diarylmethanes with 21 examples and 58–80% yields (Figure 11) [32]. Coordination of alkynes by gold catalysts triggered the attack of electron-rich benzene rings to furnish vinyl–gold intermediates (**43**) via 6-*exo*-dig cyclization. After protodeauration and [Au]^+^/H^+^ promoted isomerization, anthracenes (**45**) were obtained. An alternative pathway was also proposed; the alkyne of the substrates (**42**) was hydrolyzed under gold-catalyzed conditions to yield intermediates (**44**) that were converted to products (**45**) by an acid-mediated cyclodehydration. In addition, the products (**45**) were further modified into a variety of potentially valuable anthracene derivatives.

In 2017, a gold(I)-catalyzed tandem cycloisomerization, Diels–Alder, and retro-Diels–Alder reactions were reported by the Liu lab (Figure 12) [33]. Activation of alkyne in substrates (**46**) initiated the first cycloisomerization to yield furopyran intermediates (**47**). A subsequent Diels–Alder reaction of dienes (**47**) and dienophiles occurred to form highly strained intermediates (**48**), which underwent a retro-Diels–Alder reaction to provide biaryl products (**49**) by releasing acetaldehyde (HCHO). The above pathways were reasonably explained by density functional theory (DFT).

In 2021, the Hashmi group reported the syntheses of *meta*- and *para*dihydroxynaphthalenes based on diazoalkynes through a regiodivergent gold-catalyzed cyclization (Figure 13) [34]. The activated triple bonds of substrates (**50**) were attacked by diazocarbon to generate intermediates (**51**), followed by the formation of quinoid gold carbene intermediates (**52**) via the release of nitrogen. At this stage, two different reaction paths occurred via the addition of water or Et_3_N(HF)_3_. Under the condition of water as an additive (path a), *meta*dihydroxynaphthalenes (**54**) were produced via carbene insertion of water after protodeauration. When H_2_O and Et_3_N(HF)_3_ were used as additives, *para*dihydroxynaphthalenes (**56**) were obtained via Michael-type addition of quinoid carbene species, 1,2-phenyl migration, and protodeauration. Moreover, when only Et_3_N(HF)_3_ was used as an additive, “F^−^” was inserted instead of water for gold carbene to generate the α-fluoronaphthalenes.

Gold(I)-catalyzed arylalkyne annulations provide abundant strategies for the syntheses of benzene derivatives, including the strategies shown in this chapter and several other intramolecular or intermolecular strategies [35,36,37,38,39,40,41,42,43,44,45,46,47].

## 3. Construction of Cyclopentene Derivatives

Small-molecule skeletons containing cyclopentene are important components of many natural products and pharmaceutical intermediates. The syntheses of useful cyclopentene derivatives have attracted a great deal of interest among chemists. Gold(I)-catalyzed annulations of a variety of arylacetylene substrates provide a range of versatile synthetic methods for the syntheses of benzocyclopentene derivatives.

### 3.1. Arene-Diyne Substrates

In 2012, the Hashmi group achieved the preparation of benzofulvene derivatives based on their previous strategy of double activation of diynes containing terminal alkynes (Figure 14) [48]. Under the catalysis of a gold catalyst, dual σ/π-activated intermediates (**58**) were formed, which were rapidly transformed into gold vinylidenes (**59**) as a result of double activation. A *1,5-H* shift to electrophilic vinylidene carbon occurred, leading to intermediates (**60**). After the trapping of the carbocation by the vinyl–gold species, benzofulvene products (**61**) were synthesized in association with the elimination of the gold catalyst. The applicability of the strategy was examined by 10 examples and up to 92% yield. The above strategy was characterized by easy preparation of the substrate and a novel reaction mechanism.

In 2017, the Hashmi group the construction of aryl-substituted dibenzopentalene derivatives using terminally aromatic substituted 1,5-diyne substrates under gold-catalyzed conditions (Figure 15) [49]. One of the triple bonds was coordinated by [Au]^+^, resulting in the attack of another electron-rich triple bond to form vinyl cation intermediates (**63**). The vinyl cation was trapped by the neighboring electron-rich benzene to produce intermediates (**64**), followed by rearomatization and protodeauration to yield intermediates (**65**). Ultimately, dibenzopentalene products (**66**) were obtained by ligand exchange of gold species. It is worth noting that benzene as a solvent was not involved in the above process to trap the vinyl cation.

In 2021, the Hashmi group developed a gold-catalyzed cycloisomerization of substituted 1,5-diynes to synthesize indeno[1,2-*c*]furan derivatives. The functional group tolerance was systematically examined using 29 examples with 16–81% yields (Figure 16) [50]. Vinyl cationic intermediates (**68**) were formed through similar paths a those described previously in Figure 14 and Figure 15. Subsequently, a second annulation occurred immediately to yield oxonium intermediates (**69**). Intermediates (**71**) were produced via the release of benzyl carbocation, followed by [5,5]-sigmatropic rearrangement. Finally, indeno[1,2-*c*]furan derivatives (**73**) were obtained by rearomatization, the elimination of gold species, and proton transfer mediated by *p*-toluenesulfonic acid (PTSA). The authors fully explained the above reaction mechanism using DFT calculations, and the high regioselectivity of [5,5]-sigmatropic rearrangement was also reasonably illustrated.

### 3.2. Arene-Enyne Substrates

In 2016, Sanz et al., reported a gold(I)-catalyzed tandem reaction using *β,β*-diaryl-*o*-(alkynyl)styrenes to synthesize dihydroindeno[2,1-*a*]indene derivatives (Figure 17) [51]. A 5-*endo*-cyclization was induced by the activation of [Au]^+^ to the alkyne, which produced carbocationic intermediates (**75**). After proton elimination and protodeauration, benzofulvene intermediates (**77**) were generated. The diene units in intermediates (**77**) were then activated by the gold species to generate allylic carbocationic intermediates (**78**), which were trapped by the neighboring electron-rich aryl group to access products (**79**). In addition, under the condition of 0 °C in DCM, benzofulvene intermediates (**77**) were isolated as products.

In 2022, the Sanz lab disclosed a gold-catalyzed domino method for the syntheses of indeno[2,1-*b*]thiochromene derivatives with 21 examples and 70–88% yields (Figure 18) [52]. Activation of S/Se-substituted alkynes by [Au]^+^ triggered the cyclization of alkene to afford cationic intermediates (**81**), the carbocations of which were trapped by the vinyl–gold to produce cyclopropyl gold carbenes (**82**). The cyclopropanes of **82** were attacked by electron-rich aromatic groups to form ring-opening intermediates (**83**) after rearomatization. Indeno[2,1-*b*]thiochromene derivatives (**84**) containing sulfur or selenium were ultimately obtained by protodeauration. Importantly, when *(S)*-DM-SEGPHOS(AuCl)_2_, a chiral ligand, was used in the gold-catalyzed reaction, an enantioselective transformation was achieved in up to 80% *ee*.

### 3.3. Aryne-Enolether Substrates

In 2017, a strategy of synthesizing indeno[1,2-*c*]quinoline derivatives was developed by the Liu group through gold(I)-catalyzed cascade cyclization with 18 examples and up to 99% yield (Figure 19) [29]. The coordination of gold species to the *β* position of the triple bond initiated an attack of the enolether to generate indene intermediates (**86**). Intermediates (**87**) were produced by the activation of double bonds in the conjugated enolether with [Au]^+^, which were converted to aromatic intermediates (**88**) via intramolecular condensation with the release of MeOH after protodemetalation. In oxygen, the intermediates (**88**) were further oxidized to a more stable indeno[1,2-*c*]quinoline product (**89**). Notably, the electron-donating effect of the amine on the right benzene ring played a crucial role in the initiation of the above transformation.

In 2018, Liu et al., used 1,5-enyne substrates to synthesize a series of 2-aryl indenone derivatives in the catalysis of a gold catalyst (Figure 20) [53]. Intermediates (**92**) were formed via a gold-catalyzed cycloisomerization. An O_2_-mediated radical addition to intermediates **92** afforded intermediates (**93**) that underwent aromatization to yield peroxy intermediates (**94**), which were subsequently transformed into oxonium intermediates (**95**) through the cleavage of the peroxide bond with the coordination of [Au]^+^. Finally, indenone products (**96**) were achieved by the hydrolysis of oxonium with the release of MeOH. The above free radical reaction process was verified via control experiments and heavy atom labeling.

### 3.4. Aryne-Acetal Substrates

Arylalkynes containing acetal moieties as useful building blocks exhibited excellent reactivity in the gold-catalyzed syntheses of cyclopentene derivatives. In 2013, in pioneering work, the Toste group developed a gold-catalyzed strategy for the enantioselective syntheses of *β*-alkoxy indanone derivatives using this kind of substrate (Figure 21) [54]. It was proposed that the activation of triple bonds by gold complexes triggered the migration of an alkoxy group to the alkyne, generating oxonium intermediates (**99**) via intermediates (**98**). An enantioselective annulation then occurred to form oxonium intermediates (**100**), which were transformed into products (**101**) after isomerization. The use of [Au]^+^ with chiral ligands ensured enantioselective cyclization with up to 98% *ee*. In addition, the *β*-alkoxy indanone derivatives could be further hydrolyzed to corresponding 3-methoxycyclopentenone derivatives under PTSA conditions with wet DCM.

In 2016, Liu et al., described a gold(I)-catalyzed hydrogen-bond-regulated tandem cyclization for the syntheses of indeno-chromen-4-one and indeno-quinolin-4-one derivatives by introducing a Michael acceptor in the substrates (Figure 22) [55]. The double activation of a hydrogen bond and gold catalyst promoted methoxy migration to generate vinyl–gold intermediates (**103**), followed by an intramolecular annulation to produce intermediates (**104**) after isomerization. With conformational changing, intramolecular Michael addition occurred to yield indeno-chromen-4-one or indeno-quinolin-4-one derivatives (**105**) after the elimination of alkoxy groups.

In 2020, Sajiki and colleagues developed a gold(I)-catalyzed approach for the preparation of indenone derivatives using arylalkyne substrates containing cyclic acetals (Figure 23) [56]. The triple bonds were first activated by the gold complex to produce vinyl–gold intermediates (**107**), which initiated the migration of benzylic hydride to generate oxonium cationic intermediates (**108**). Cyclized gold(I)–carbene intermediates (**109**) were then formed by intramolecular nucleophilic addition. At this stage, when the system contained water, a carbene insert process occurred to yield intermediates (**110**), followed by a [Au]^+^-activated dehydration reaction to produce indenone derivatives (**112**). Alternatively, products (**112**) were generated directly from the cyclized gold(I)–carbene intermediates (**109**) through a *1,2-H* shift and elimination of gold species. The key 1,5-hydride shift was verified by deuterium-labeled experiments and 2D NMR analysis.

In 2020, the Liu group reported a gold(I)-catalyzed domino reaction to construct benzo[*b*]indeno[1,2-*e*][1,4]diazepine derivatives using *o*-phenylenediamines and ynones (Figure 24) [57]. The coordination of the gold species with a triple bond induced a series of transformations into intermediates (**115**), which was similar to the generation of intermediates (**104**) shown in Figure 22. The intermediates (**115**) underwent Michael addition with exogenous *o*-phenylenediamine to produce intermediates (**117**) after the elimination of MeOH. Ultimately, benzo[*b*]indeno[1,2-*e*][1,4]diazepine derivatives (**118**) were synthesized by intramolecular condensation and aromatization accompanied by the elimination of MeOH and H_2_O. Controlled experiments were further conducted to determine the rationality of the above reaction.

Recently, the Liu group developed a synthetic strategy for 2,2′-spirobi[indene] derivatives using arylalkyne–acetal substrates based on their previous research (Figure 22 and Figure 24), mainly involving methoxylation and aldol condensation (Figure 25) [58]. Intermediates (**120**) were easily produced by the activation of [Au]^+^/H^+^ and converted into intermediates (**121**) through an intramolecular aldol reaction. After releasing MeOH, 2,2′-spirobi[indene] derivatives were obtained. It should be noted that the reversible equilibrium of aldol/retro-aldol reactions led to the isomerization of the hydroxyl group.

### 3.5. Other Arylalkyne Substrates

In 2021, Xu and colleagues achieved a cascade strategy for the syntheses of indene derivatives involving gold(I)-catalyzed Wolff rearrangement and ketene C=C dual functionalization (Figure 26) [59]. Diazoketone substrates (**123**) were activated by a gold complex to form gold carbine, which as converted to ketene intermediates (**124**) by Wolff rearrangement. The ketene units of **124** were then attacked by nucleophiles (ROH) to form enol intermediates (**125**). Activation of a triple bond by gold(I) species initiated a C-5-*endo*-dig cyclization to obtain indene products (**126**). In addition, when nucleophiles such as indoles or pyrroles were used, O-7-*endo*-dig cyclization occurred to generate benzo[*d*]oxepine derivatives. The scope of the above strategy was studied in detail with 46 examples and up to 88% yield, and the related reaction pathways were explained by DFT calculations.

In 2021, a strategy for the syntheses of indene derivatives based on the cyclization of ynamides was developed by the Evano group (Figure 27) [60]. Gold–keteniminium ions (**128**) were formed upon the coordination of [Au]^+^ to the triple bond in ynamide, followed by a *1,5-H* shift, resulting in carbocation intermediates (**129**). Subsequently, the carbocations of **129** were trapped by vinyl–gold to trigger a cyclization, producing intermediates (**130**). After a *1,2-H* shift and elimination of [Au]^+^, indene products (**131**) were achieved. Alternatively, indene products (**131**) could be formed by the elimination of a proton and protodeauration. This method is associated with a wide range of substrates and was systematically studied using 20 examples with 40–96% yields.

Recently, the Ohno lab reported a gold(I)-catalyzed cascade acetylenic Schmidt reaction/*1,5-H* shift/*N*- or *C*-cyclization method producing indole[*a*]- and [*b*]-fused polycycle derivatives (Figure 28) [61]. The isotopic labeling experiment showed that the reaction started with an acetylenic Schmidt reaction activated by gold species, which resulted in the formation of *α*-imino gold carbenes (**133**), followed by a *1,5-H* shift to yield carbocationic intermediates (**134**), which were in reversible equilibrium with aromatized intermediates (**135** and **137**). Finally, *C*-cyclization products (**136**) were generated via aromatized intermediates (**135**), and the *N*-cyclization products (13b) were yielded via aromatized intermediates (**137**) with a bond rotation. Notably, the selectivity of the *N* and *C*-cyclization products could be tuned through the electron density of the left benzene ring, the stability of the carbocation, and the effect of the counterion. Moreover, the above strategy is excellent example of benzylation of benzylic C(sp^3^)-H functionalizations, providing a concise method for the syntheses of indole[*a*]- and [*b*]-fused polycycle derivatives.

Based on the cases summarized in this chapter, it seems that the gold(I)-catalyzed tandem approach using a variety of arylalkyne substrates could be used to synthesize corresponding cyclopentene derivatives, such as benzofulvenes, dibenzopentalenes, 2,2′-spirobi[indene], indenes, etc. These structurally diverse cyclopentene derivatives can provide further possibilities for the discovery of bioactive lead compounds and provide strategic support for the syntheses of related bioactive molecules.

## 4. Construction of Furan and Pyran Derivatives

Furan and pyran derivatives are valuable heterocyclic skeletons and important intermediates for the syntheses of drugs and lead compounds. For example, benzofuran derivatives exhibited excellent inhibition of both drug-sensitive and drug-resistant pathogens through a unique antitubercular and antibacterial mechanism [62]. Gold(I)-catalyzed arylalkyne cyclization can be used to construct a variety of furan- or pyran-containing derivatives, such as polycyclic furans, polycyclic pyrans, benzofurans, and benzopyrans. In this chapter, we discuss in detail the synthetic strategies and the scope of furan and pyran derivatives depending on the arylene substrates.

### 4.1. 1,5-Enyne Substrates

1,5-enyne is an important building block in the gold(I)-catalyzed construction of small-molecule heterocycles. In 2016, Liu and colleagues reported a gold(I)-catalyzed tandem strategy for the syntheses of furopyran derivatives involving Claisen rearrangement and 6-*endo*-trig cyclization, the regioselectivity of which was mainly controlled by the angle strain of propargyl *γ*-butyrolactone-2-enol ethers (**139**) (Figure 29) [63]. A 6-*endo*-dig cyclization was initiated by the coordination of the gold catalyst to the triple bond to form intermediates (**140**) that were rearranged into *β*-allenic ketones (**141**). Intermediates (**143**) were produced by keto–enol tautomerism, and angle strain controlled 6-*endo*-dig cyclization. After demetallation, furopyran derivatives (**144**) were successfully obtained. The reason for the change in regioselectivity from 5-*exo*-trig to 6-*endo*-trig was explained by DFT calculation.

In the same year, Liu and colleagues achieved the syntheses of multisubstituted furofuran derivatives based on the studies represented in Figure 29 by trapping key intermediates (**141**) (Figure 30a) [64]. Alkynes of substrates (**145**) were activated by gold species to induce a rearrangement reaction and yield allene intermediates (**147**), consistent with the generation of intermediates (**141**). The terminal alkene of the allene was coordinated by a [Au]^+^ complex to enable the attack of nucleophiles, generating *σ*-allyl gold species (**148**). After SE’-type protodeauration of **148**, intermediates (**149**) were accessed, the enolether units of which were activated by gold species to trigger a 5-*exo*-trig cyclization. Finally, furofuran products (**150**) were delivered after protodeauration. In addition, multisubstituted furopyran derivatives were successfully produced when the propargyl terminal was substituted with thiophene or furan (Figure 30b). Substrates (**151**) were converted to intermediates (**152**) under the activation of a gold catalyst, which was similar to the formation of intermediates (**140**) shown in Figure 30. Intermediates (**152**) were not rearranged to *β*-allenic ketones due to the chelation of the heteroatom to the gold complex but were transformed to intermediates (**153**). Ultimately, furopyran products (**154**) were obtained via the nucleophilic addition of oxonium moiety after protodeauration. Thus, the authors achieved the syntheses of furofuran and furopyran derivatives by substituent modulation using propargyl vinyl ethers in the catalysis of gold(I) catalysts.

In 2016, Jiang et al., developed a gold(I)-catalyzed, ligand-regulated cyclization for the syntheses furopyran or dihydroquinoline derivatives using 1,5-enyne substrates containing directing groups (Figure 31) [65]. When using tris(2,4-di-*tert*-butylphenyl) phosphite (L1) in combination with trifluoromethanesulfonate (OTf^−^), gold(I)–π-alkyne intermediates (**156**) were formed by three coordinations, which were attributed to the increased electrophilicity of the gold center. The activated triple bond was attacked by the ortho position of the left aromatic ring, which overcame the steric hindrance. After protodeauration, furopyran and dihydroquinoline derivatives (**158**) were accessed (Figure 31a). When a combination of Xphos ligand (L2) and NTf_2_^−^ was used, intermediates (**160**) were produced, which were attributed to the decreased electrophilicity of the gold center. Next, the activated alkyne was attacked by the para position of the left aromatic ring, which yielded products (**162**) after protodeauration (Figure 31b). The above regiodivergent cyclization depended mainly on the electronic and steric effects of the ligand in gold species. The authors systematically examined the scope of the above switchable strategy with moderate to excellent yields.

### 4.2. Alkyne–Phenol Substrates

In 2016, a gold-catalyzed tandem cyclization to benzofuran derivatives was reported by Saito and colleagues (Figure 32) [66]. The coordination of a gold complex to the triple bond initiated cyclization to generate intermediates (**164**), which were subsequently transformed into intermediates (**165**) with an *α*-alkoxy alkyl-shift. Under the influence of the activation of a gold catalyst, oxonium intermediates (**166**) were formed by releasing R^2^OH, the *α,β*-enone moieties of which were attacked by the nucleophilic group to generate benzofuran products (**167**). Moreover, this strategy could be used for the construction of a larger number of small-molecule heterocyclic derivatives by regulating side chains in *o*-alkyl aryl ethers (**163**).

In 2019, the González lab achieved a gold(I)-catalyzed tandem cycloisomerization for the syntheses of benzofuran derivatives using 2-(iodoethynyl)-aryl esters with 15 examples and up to 85% yield (Figure 33) [67]. The triple bonds of substrates (**167**) were activated by the gold complex to generate gold–vinylidene intermediates (**168**) via a 1,2-iodine shift. 3-iodo-2-acyl benzofuran products (**169**) were assembled by inserting gold carbine into the O-acyl bond. Importantly, the capture of intermediates (**168**) by silane through supplementary experiments implied a gold-catalyzed iodine rearrangement.

A series of vinyl benzofuran derivatives was synthesized via a gold(I)-catalyzed cascade cyclization/hydroarylation method developed by the Xia group in 2022 (Figure 34) [68]. With SIPrAuCl as catalyst and NaBARF as cocatalyst, benzofurans (**171**) were formed from *o*-alkyl phenol substrates (**170**). The triple bond of the haloalkyne was activated by the gold complex and thus attacked by the C3 position of the benzofuran through transition states (**172**). Then, cationic vinyl–gold intermediates were produced, which were then transformed into vinyl benzofurans (**173**) through a proton transfer. The authors demonstrated the reaction mechanism via experiments and computational calculations, and the functional group tolerance of the above strategy was examined with 20 examples and 19–98% yields.

### 4.3. Other Arylalkyne Substrates

In 2018, the Xu group synthesized furan derivatives using a series of propargyl diazoacetates through a gold(I)-catalyzed, water-involved tandem approach with 29 examples and up to 90% yield (Figure 35) [69]. Initially, diazoacetate substrates (**174**) were transformed into gold carbene intermediates via the activation of the gold catalyst with the release of N_2_, the gold carbene moieties of which were then attacked by H_2_O to form oxonium ylides (**175**). After isomerization, enol intermediates (**176**) were produced by proton transfer, followed by a 6-*endo*-dig cyclization to yield cyclized intermediates (**177**). The carbonyl groups of **177** were nucleophilically attacked by the vinyl–gold to generate ring contraction intermediates (**178**). After the cleavage of cyclopropane, secondary carbene intermediates (**179**) were generated with the elimination of H_2_O via an intramolecular H-bond-assisted pinacol rearrangement. When R^2^ or R^3^ was H, the final processes of *β*-H elimination and protodeauration yielded furan products (**180**). The authors demonstrated the formation of intermediates by interception experiments and verified the involvement of H_2_O by isotope-labeled experiments.

In the same year, Liu and colleagues reported a gold(I)-catalyzed tandem protocol involving oxidation, 1,2-enynyl migration, and 6-*exo*-dig cyclization to prepare 1*H*-isochromene derivatives (Figure 36) [70]. The R^3^-substituted alkyne of o-(alkynyl)-phenyl propargyl ether substrates (**181**) was coordinated by the gold catalyst to initiate an attack of *N*-oxide, followed by the elimination of the pyridine derivative to generate gold carbene intermediates (**182**). Next, a novel 1,2-enynyl migration resulted in the formation of oxonium ion intermediates (**183**), which were then converted into 1*H*-isochromene products (**184**) by 6-*exo*-dig cyclization after protodeauration. Notably, the reaction mechanism was supported by isotopic labeling experiments.

There are many excellent examples of the syntheses of furan and pyran derivatives reported, other than those listed in this chapter [71,72,73,74,75], including multicomponent, one-pot reactions [76,77]. Gold(I)-catalyzed tandem reactions are significant for the construction of small-molecule scaffolds containing furan or pyran. Furthermore, the development of gold(I)-catalyzed strategies also provides material support for the study of the bioactivity of furan and pyran derivatives.

In addition, the use of gold(I)-catalyzed alkyne cyclization to construct *N*-heterocyclic skeletons, e.g., pyrrole, indole, quinoline, pyridine, carbazole, is an important research direction. This class of reactions has been systematically summarized in recent reviews, so is not be described repeatedly in this feature paper [78,79,80].

## 5. Conclusions

In the last decade, homogeneous gold(I)-catalyzed cyclization for the construction of small-molecule skeletons from arylalkyne substrates has been developed rapidly, owing to the ease of substrate preparation and the stability of gold catalysts.

In this feature paper, we systematically summarized the gold(I)-catalyzed syntheses of benzene, cyclopentene, furan, and pyran derivatives, which were carefully classified according to the type of arylalkyne substrate. Gold(I)-catalyzed tandem approaches for the construction of small-molecule scaffolds generally involve cyclization, isomerization, aromatization, migration, rearrangement, and other processes that are usually verified by controlled experiments and isotopic labeling experiments, as well as DFT calculations. In addition, the efficient strategies of gold catalysis were featured, with good functional group tolerance and reaction yield.

Although many excellent works have been reported with respect to gold catalysis for the syntheses of small-molecule skeletons, additional gold(I)-catalyzed asymmetric strategies are urgently required. Therefore, studies on the enantioselective construction of small-molecule scaffolds with the participation of chiral ligands will be further developed. In addition, dual gold/photoredox-catalyzed or dual gold/enzyme-catalyzed organic reactions can contribute to the development of this field [81].

## Data Availability

Not applicable.

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
