# Peer review of "Recent Advances in Gold(I)-Catalyzed Approaches to Three-Type Small-Molecule Scaffolds via Arylalkyne Activation"

_molecules, 2022, doi:10.3390/molecules27248956_

Round 1

Reviewer 1 Report

Yang and Liu et al. reported ‘’ Recent advances in Gold(I)-catalyzed approaches to three-type small molecules scaffolds via arylalkyne activation’’.  In this manuscript, the author summarized the wide scope of gold-catalyzed reaction for construction of benzene derivatives, cyclopentene derivatives and furan/pyran derivatives.  This manuscript is very useful for many readers who want to know or start gold catalyzed reaction. So, the reviewer recommends its publication of this work in ‘’Molecules’’ after minor revision as follows.

Page 4, Scheme 4,

Correction to extend the arrow from the oxygen of the pyridine oxide.

Page 7, Scheme 9

The orientation of XH is wrong.

It looks as if it is bounded to H from the aromatic ring.

Correct to HX instead of XH.

Page 11, Scheme 18

X=S in scheme is wrong.

Half-width spaces should be placed on both sides of =.

Ex. X = S, Se

Author Response

To Reviewer 1:

Question 1

Page 4, Scheme 4. Correction to extend the arrow from the oxygen of the pyridine oxide.

Response:

We are appreciated for your advice. The arrow was corrected to extend from the oxygen of the pyridine oxide in the revised manuscript.

Question 2

Page 7, Scheme 9. The orientation of XH is wrong. It looks as if it is bounded to H from the aromatic ring. Correct to HX instead of XH.

Response:

Thanks for the good advice, XH has been replaced with HX in our revised manuscript.

Question 3:

Page 11, Scheme 18. X=S in scheme is wrong. Half-width spaces should be placed on both sides of =. Ex. X = S, Se

Response:

Thanks for your advice, formatting was corrected in the revised manuscript

Reviewer 2 Report

This review article by Yongxiang Liu et. al. was well written describing the recent advances in gold(I)-catalyzed approaches to three type small molecule scaffolds via arylalkyne activation. The review article was well framed describing the synthesis of various benzene, cyclopentene, furan and pyre derivatives. Even though a compile of examples describing the synthesis of indole and carbazoles is described a separate section related to synthesis of azaheterocycles (pyrrole, indole, quinoline, pyridine, carbazole and other) would add more impact to the review article. For example the following references might be helpful for for the construction of carbazole and spirooxindole. 

DOI: 10.1039/C6CC00633G for carbazole synthesis
DOI: 10.1039/C6QO00430J for spirooxindole synthesis

Author Response

To Reviewer 2:

Question 1:

Even though a compile of examples describing the synthesis of indole and carbazoles is described a separate section related to synthesis of azaheterocycles (pyrrole, indole, quinoline, pyridine, carbazole and other) would add more impact to the review article.

Response:

Thanks for the good advice. “In addition, gold(I)-catalyzed alkyne cyclization to construct N-heterocyclic skeletons, e.g. pyrrole, indole, quinoline, pyridine, carbazole and so on, is also an important research content. However, this class of reactions has been systematically summarized in recent reviews, which will not be described repeatedly in this feature paper [78-80]” have been added in our new manuscript.

Question 2:

For example, the following references might be helpful for for the construction of carbazole and spirooxindole. DOI: 10.1039/C6CC00633G for carbazole synthesis. DOI: 10.1039/C6QO00430J for spirooxindole synthesis.

Response:

We are appreciated for your advice. After thorough examination, it was found that these two articles on silver catalysis are out of the theme of this paper on gold catalysis. Accordingly, the latest reviews on the construction of N-heterocyclic skeletons have been added in the revised manuscript (DOI: 10.1021/acs.chemrev.0c00788; 10.1002/tcr.202100159; 10.1002/chem.202002154).

Round 2

Reviewer 2 Report

Paper can be accepted after checking for English language and spell checks.